# Kinematic Evidence of Root-to-Shoot Signaling for the Coding of Support Thickness in Pea Plants

**DOI:** 10.3390/biology11030405

**Published:** 2022-03-06

**Authors:** Silvia Guerra, Bianca Bonato, Qiuran Wang, Alessandro Peressotti, Francesca Peressotti, Walter Baccinelli, Maria Bulgheroni, Umberto Castiello

**Affiliations:** 1Department of General Psychology, University of Padova, 35131 Padova, Italy; silvia.guerra.5@phd.unipd.it (S.G.); bianca.bonato.1@phd.unipd.it (B.B.); qiuran.wang@phd.unipd.it (Q.W.); 2Department of Agricultural, Food, Environmental and Animal Sciences, University of Udine, 33100 Udine, Italy; alessandro.peressotti@uniud.it; 3Department of Developmental Psychology and Socialization, University of Padova, 35131 Padova, Italy; francesca.peressotti@unipd.it; 4Ab.Acus S.r.l., 20155 Milan, Italy; walterbaccinelli@ab-acus.eu (W.B.); mariabulgheroni@ab-acus.com (M.B.)

**Keywords:** plant behavior, root-to-shoot signaling, circumnutation, climbing plants, kinematics

## Abstract

**Simple Summary:**

A growing body of evidence has reported that climbing plants show the unique ability of being able to locate a support in the environment and recognize some of its features in order to modulate their searching and attachment behavior accordingly. To date, the mechanism underlying the coding of a support’s thickness is yet to be uncovered. Here, we investigate the interaction between the above- (i.e., stem and tendrils) and below-ground (i.e., the root system) organs of pea plants (*Pisum sativum* L.) to gain information about a support and to modulate their behavior towards it. The results suggest that the coding of a support’s thickness is achieved via a functional equilibrium subtended by crosstalk between the grounded and the aerial components of the plant.

**Abstract:**

Plants such as climbers characterized by stems or tendrils need to find a potential support (e.g., pole, stick, other plants or trees) to reach greater light exposure. Since the time when Darwin carried out research on climbing plants, several studies on plants’ searching and attachment behaviors have demonstrated their unique ability to process some features of a support to modulate their movements accordingly. Nevertheless, the strategies underlying this ability have yet to be uncovered. The present research tries to fill this gap by investigating how the interaction between above- (i.e., stems, tendrils, …) and below-ground (i.e., the root system) plant organs influences the kinematics of their approach-to-grasp movements. Using three-dimensional (3D) kinematic analysis, we characterized the movements of pea plants (*Pisum sativum* L.) as they leaned towards supports whose below- and above-ground parts were characterized by different thicknesses (i.e., thin below- thick above-ground, or the opposite). As a control condition, the plants were placed next to supports with the same thickness below and above ground (i.e., either entirely thin or thick). The results suggest that the information regarding below- and above-ground parts of a support appears to be integrated and modulates the reach-to-grasp behavior of the plant. Information about the support conveyed by the root system seems to be particularly important to achieve the end-goal of movement.

## 1. Introduction

Climbing plants need to attach themselves to external supports to grow vertically in order to enhance their light acquisition. The availability of supports influences climber diversity in a variety of environments, and it is well established that climbers that fail to encounter a support often show reduced growth and/or reproduction compared with those successfully climbing up an external support [1]. This has been observed in forests [2], open habitats [3] and controlled environments [4]. Support-finding not only involves enhanced fitness, but also triggers changes in growth form, biomass allocation, morphology, and physiology [4,5,6,7,8]. Locating a suitable support seems thus to be a key process in the life history of climbing plants. 

The study of climbing plants’ behavior began with Darwin’s observations regarding helical organ movement, which was exploratory in nature and performed by the stem and the tendrils; he named it circumnutation [9]. Since then, a variety of studies have investigated the mechanisms underlying the searching and attachment behaviors of climbing plants at different levels [10,11,12,13,14,15].

However, despite the large number of works appearing in the literature, little is known regarding how climbing plants decide how to search for and attach themselves to supports [3]. In this respect, Darwin [9] conducted simple indoor experiments, noticing that vines were not only able to locate supports and lean towards them, but they also, in some circumstances, could show averse behavior towards them. This effect was first described for *Bignonia capreolata* L. tendrils. When the tendrils of that plant encountered a support (i.e., a stick), they bent and curled around it. Instead, when a support was perceived as ‘inadequate’ in terms of thickness or excessive smoothness, the tendrils let go of it. A similar phenomenon was observed when herbaceous twining vines encountered a very thick tree trunk. Instead of winding around it, they wound around themselves. Darwin [9] remarked that it would have been maladaptive for the vines to wrap themselves around thick, trees, as it was unlikely that they would reach higher light levels by the end of a single growing season.

The cases cited above provide some support to the claims that climbing plants can modify their circumnutation patterns to a greater extent with respect to what would be expected by chance alone depending on some features of the targeted support [6]. Indeed, several attributes of a host tree seem to determine the probability of being colonized by some climbers [3]. The thickness (i.e., diameter) of a support does seem to affect their suitability. Both theoretical and empirical approaches have shown that when the diameter of the support increases beyond a certain point, twining plants are unable to maintain tensional forces and they become unattached [13,16]. The fact that these plants have problems in curling around a thick support was already noted by Darwin [9], who reported that the shoots of a *Wisteria sinensis* (Sims) sweet twining vine were unable to climb up a support that was about 15 cm wide. Field studies in tropical, sub-tropical and temperate rainforests have confirmed that the relative abundance of stem twiners decreases with increasing tree diameter [17,18]. The support-size biomechanical constraints for twining plants are intermediate compared with tendril climbers, whose upper limit of usable trunk diameter is even lower [13,19]. 

Evidence from laboratory settings has further elucidated the effects of the thickness of a support by using kinematics to characterize the movements of the *Pisum sativum* L. (from now on *P. sativum*) [20,21,22,23,24]. Guerra and colleagues [22], for example, demonstrated that the *P. sativum* plant can perceive a support and modulate the kinematics of the tendrils’ velocity and aperture depending on the thickness of the support. Both the average and the maximum velocity of the tendrils were found to be higher for thinner with respect to thicker supports. In addition, it took more time for the tendrils to reach peak velocity and the maximum aperture when the support was thinner. Likewise, the maximum distance between the tendrils is significantly greater for thinner with respect to the thicker supports [20,21,22,23]. This phenomenon can be explained by the fact that reaching and grasping thicker supports is more energy-consuming for plants than grasping thinner ones. Indeed, grasping a thick support implies that the plant must increase the length of its tendrils in order to efficiently curl around it [25], and it also needs to strengthen the tensional forces in order to resist the force of gravity [6]. Since these processes are characterized by a high consumption of adenosine triphosphate (ATP), the energy expenditure is higher for coiling around thicker supports [13]. The reduction in movement velocity during an approaching maneuver and a greater aperture may allow climbing plants to preserve energy for the coiling phase in order to reduce the risk of error and to facilitate the movements needed to securely attach themselves to a support. The reduction in movement velocity may also serve to lengthen the time window within which tendrils can establish contact points with the support. The extra time needed to reach a thick support may allow climbing plants to correct their trajectories and to select with greater precision the contact points for efficiently establishing a firm hold of the support.

Although both ecological and laboratory findings have provided data regarding the relationship between the behavior of climbers and support thickness, the mechanisms underlying thickness coding and processing remain obscure [20,21,22,23]. A variety of hypotheses based on the perceptual abilities of plants have been put forward [26]. For example, it has been advanced that proprioception may allow climbing plants to perceive the position of their tendrils, which provides the feedback information necessary for adjusting their aperture to the thickness of the support [27,28]. In addition, plants may have at their disposal some systems typically involved in the reception of light and the capacity to build a representation of the surrounding environment [29,30]. Continuing this analysis, recent studies report that plants may get information about their surroundings by emitting sonic clicks (i.e., clicking) and perceiving the returning echoes [31]. Finally, plants may process external stimuli using chemoreception of volatiles. Evidence suggests that plants release airborne chemicals that can convey ecologically relevant information about the stimuli with which they are interacting [32,33].

While the sensory mechanisms of the aerial sector of climbing plants are at work checking out the features of a support, the extreme tips of the roots (i.e., root cap) may also be involved in thickness sensing. Indeed, the root caps respond to numerous signals (e.g., gravity, touch, humidity) and assess and dynamically control the direction of root growth [34,35,36,37]. A great deal of plant decision-making can be ascribed to the root cap system [38]: consider the highly sophisticated responses of roots, such as gravitropism and thigmotropism [39,40]. Indeed, roots stop developing downwards when they encounter a physical obstacle and instead begin to grow horizontally [41]. The touch sensor might be the most likely candidate in obstacle perception given its immediate physical interaction with the obstacle. However, there is also evidence that obstacle avoidance may rely on root exudates, the cocktail of compounds secreted by roots allowing the plants to explore the soil environment and to gain information from it [42,43]. In fact, root recognition and navigation around physical obstacles is prevented when activated charcoal, which absorbs many compounds, is mixed to the substrate [44].

The pivotal importance of the root system for coding the thickness of the support has been decisively revealed in a recent study focusing on *P. sativum* plants [23]. One experiment investigated the movement of tendrils as it leaned towards a support inserted into the substrate. In another experiment, the same support was lifted out the ground to prevent the root system from sensing it. The results confirmed that the tendrils’ kinematics depended on information about the thickness of the support provided by the root system. When the support was no longer inserted into the ground, the plants continued to circumnutate but eventually fell to the ground. These results suggest that the root system can perceive the presence and the thickness of a support. Importantly, the specific information perceived by the roots affects the planning and the execution of the approach-to-grasp movements of climbing plants. They likewise suggest that the plant’s above-ground organs are unable to code for the thickness of the support without the sensory input of the roots. These intriguing and puzzling inferences inevitably lead us to ask: why do *P. sativum* plants rely on underground stimuli to drive aboveground behavior given that what the root system finds in the soil may not be a reliable proxy for what is happening above it? 

In light of all these considerations, the present study set out to further investigate the contribution of the root system to coding information about the support thickness by exploring the interactivity between the root system and the aerial parts of a climbing plant [45]. Experiments were thus designed to study the movement of *P. sativum* plants towards supports whose above-ground and underground parts had different thicknesses (i.e., perturbed conditions). A group of plants whose supports had a thin below-ground part and a thick above-ground one (‘Thin-Below’ condition; see Figure 1a) was tested. Another group with inverse conditions i.e., the support was thick below-ground and thin above-ground (‘Thick-Below’ condition; see Figure 1b) was also tested. Trials were also carried out using single-thickness supports (either thin or thick); these were considered control conditions. Specifically, trials for the ‘Thin-Below’ perturbed condition were compared with trials for the ‘Control-Thick’ condition (Figure 1a,d), and movements during the ‘Thick-Below’ condition were compared with those during the ‘Control-Thin’ condition (Figure 1b,c). 

We hypothesized that if roots play a pivotal role in sensing the support’s thickness, then kinematic parameterization would be strictly associated to the thickness of the below-ground part of the support. We would thus expect to see differences between the perturbed and the control conditions because the thickness for the below-ground part of the support used for the perturbed conditions differed from the thickness of the support used for the control conditions. Remember that the movement of *P. sativum* plants towards supports of different thicknesses are characterized by specific kinematic signatures [20,21,22,23].

On the other hand, if the aerial part of the plant is involved in sensing the support thickness, then there should be no significant differences between the perturbed and control conditions given that the thickness of the upper part of the supports used for the perturbed conditions was similar to that of the supports used for the control conditions. 

A third hypothesis postulated that a crosstalk goes on between the roots and the aerial parts of the plant that would be affected by a mismatch between the below- and aboveground part of the supports. If the below-ground information is incongruent with the end-goal of the movement, an adjustment by the above-ground organs (i.e., stem, tendrils, …) should occur and the movement adapted accordingly.

## 2. Materials and Methods

### 2.1. Subjects

Forty snow peas (*P. sativum* var. saccharatum ‘Carouby de Maussane’) were chosen as the study plants (see Table 1). The plants were randomly assigned to the experimental conditions.

### 2.2. Supports

The supports were wooden poles 54 cm high (the below-ground part of the supports were 7 cm long, while the above-ground part was 47 cm long), the supports were positioned 12 cm away from the plant’s first unifoliate leaf (Figure 2). The supports had different diameters depending on the experimental conditions (see Figure 1). For the (i) ‘Thin-below’ perturbation condition (Figure 1a) the below-ground part of the support had a diameter of 1.2 cm, and the above-ground part had a diameter of 3 cm; (ii) For the ‘Thick-below’ perturbation condition (Figure 1b), the below-ground part of the support had a diameter of 3 cm and the above-ground part had a diameter of 1.2 cm, (iii) For the ‘Control-Thin’ condition (Figure 1c) the diameter was 1.2 cm; (iv) For the ‘Control-Thick’ condition (Figure 1d) the diameter was 3 cm.

### 2.3. Growth Conditions and Germination

Cylindrical pots (diameter 20 cm; height 20 cm) were filled with silica sand (type 16SS, dimension 0.8/1.2 mm, weight 1.4). The pots were watered and fertilized using a half-strength solution culture (Murashige and Skoog Basal Salt Micronutriment Solution; 10×, liquid, plant cell culture tested; SIGMA Life Science, Milan, Italy) and then watered with tap water as needed three times a week. One seed per pot was placed 6 cm away from the pot’s border and sowed at a depth of 2.5 cm. Each pot was enclosed in a growth chamber (Cultibox SG combi 80 × 80 × 160 cm; Figure 2) so that the seeds could germinate and grow in controlled environmental conditions. The chamber air temperature was set at 26 °C; an extractor fan was equipped with a thermo-regulator (TT125; 125 mm-diameter; max 280 MC/H vents) and an input-ventilation fan (Blauberg Tubo 100–102 m^3^/h) was also operating. The two-fan combination was used to allow for a steady air flow rate in the growth chamber with a mean air residence time of 60 s. The fan was placed in such a way that the air currents did not affect the plants’ movements. The plants were grown with an 11.15-hour photoperiod (5.45 am to 5 pm) under a cool white LED lamp (V-TAC innovative LED lighting, VT-911-100W, Des Moines, IA, USA or 100 W Samsung UFO 145 lm/W—LIFUD) that was positioned 50 cm above each seedling. A LI-190R quantum sensor (Photosynthetic Photon Flux Density of 350 umol_ph_/(m^2^s; Lincoln, NE, USA) was positioned 50 cm above the seedling. Reflective Mylar^®^ film was used on the chamber walls to optimize the uniformity of the light distribution. The experimental methodology was utilized for the single plants grown individually in the growing chamber. In order to sustain the plants after germination, a wooden stick of 10 cm in height and 0.3 cm in diameter was attached to the plant (Figure 3a). The stick was built as an L shape and fixed to the pot’s rim. The part running from the vertical component of the stick to the pot’s rim was not grounded and covered by a thin layer of silica sand. Treatments were replicated five times by randomly assigning treatments to the eight growing chambers.

### 2.4. Video Recording and Data Analysis

A pair of RGB-infrared cameras (i.e., IP 2.1 Mpx outdoor varifocal IR 1080P) were placed 110 cm above the ground, spaced at 45 cm to record stereo images of the plant within each growth chamber. In accordance with the experimental protocol, a frame was synchronously acquired every 3 min (frequency 0.0056 Hz) by the cameras. Markers on the anatomical landmarks of interest, namely the tips of the tendrils were inserted post-hoc (Figure 3a). The markers were also positioned on the support (i.e., on both the lowest and the highest points of the support) as reference points (Figure 3a). An ad hoc software (Ab.Acus s.r.l., Milan, Italy) developed in Matlab was used to position the markers, track their position frame-by-frame on the images acquired by the two cameras so as to reconstruct the 3D trajectory for each marker. The tracking procedures were at first performed automatically throughout the time course of the movement sequence using the Kanade–Lucas–Tomasi (KLT) algorithm on the frames acquired by each camera after distortion removal. The tracking was manually verified by an experimenter who checked the markers’ position frame-by-frame. The 3D trajectory of each tracked marker was computed by triangulating the 2D trajectories obtained from the two cameras (Figure 3b). The cameras were connected via Ethernet cables to a 10-port wireless router (i.e., D-link Dsr-250n) connected via Wi-Fi to a PC; the frame acquisition and saving process were controlled by CamRecorder software (Ab.Acus s.r.l., Milan, Italy). To maximize the contrast between the anatomical landmarks of the *P. sativum* plants (e.g., the tendrils) and the background, black felt velvet was fixed on some sectors of the walls of the boxes and the wooden supports were darkened with charcoal. The intrinsic, extrinsic and the lens distortion parameters of each camera were estimated using a Matlab Camera Calibrator App. Depth values of the single images were calculated by taking 20 pictures of a chessboard (squares with 18 mm of side, 10 columns, 7 rows) from multiple angles and distances in natural non-direct light conditions. For stereo calibration, the chessboard used for the single camera calibration process was placed in the middle of the growth chamber. The photos were then taken by the two cameras to extract the stereo calibration parameters. The time sequence of plants’ movement was analyzed whenever a plant grasped a support. The initial frame of the sequence was defined as the frame in which the tendrils of the considered leaf were visible from the apex. The end of the time sequence was defined as the frame in which the tendrils started to coil around the support. 

On the basis of previous evidence [22,23] the dependent variables chosen to test our experimental hypothesis were: (i) the movement time (min); (ii) the spatial trajectories of the landmarks considered; (iii) the maximum velocity of the tendrils during circumnutation (mm/min); (iv) the time it took for the maximum tendril velocity to be reached as a percentage of movement duration (%); (v) the maximum aperture of the tendrils corresponding to the maximum distance reached by the tip of the tendrils during the approach phase (mm); (vi) the time it took for the maximum aperture of the tendrils to be reached as a percentage of movement duration (%); (vii) the maximum duration of the circumnutation (min) and (viii) the maximum length of the circumnutation (mm).

The median values of each of the dependent measures considered were compared across all the conditions using the Wilcoxon rank-sum test (one-tailed). In addition to the W-statistic and the *p*-value, we also analyzed the size of the effect calculated as r = z/√N, in which z is the z-score and N is the total number of observations [46]. All statistical analyses were carried out using computing environment R [47] software and the wilcox.test function.

## 3. Results

### 3.1. Qualitative Results

The tip of the tendrils showed a movement pattern that could be defined as circumnutation during all of the experimental conditions (Figure 3a,b); circumnutation, as explained earlier, is a helical organ movement which should help climbers find suitable supports in the environment. Once the plant detected and perceived a support, it strategically modified the trajectory of its tendrils and began to bend towards it to approach and clasp it. Importantly, the plants directed their movement toward the support and shaped the choreography of the tendrils depending on the thickness of the support even before any physical contact was made [20,21,22,23] (see Appendix A Video S1 and S2).

### 3.2. Kinematic Results

#### 3.2.1. Control Conditions

The overall picture of the results for the control conditions mirror those reported in previous studies in which ‘thick’ and ‘thin’ supports were compared [20,21,22,23]. Movement time was longer for the ‘Control-Thick’ than the ‘Control-Thin’ condition (1638 vs. 1145 min; W = 333; *p* = 0.042; r = 0.26; Figure 4).

The maximum velocity of the tendrils was lower (12 vs. 14 mm/min; W = 208; *p* = 0.023; r = 0.29; Figure 5a) and the time it occurred was earlier for the thicker than for the thinner supports (67 vs. 81%; W = 400; *p* = 0.041; r = 0.26; Figure 5a). The maximum aperture of the tendrils was wider (39 vs. 30 mm; W = 114; *p* = 0.008; r = 0.34; Figure 5b) and the time at which it occurred was later for the thinner than for the thicker supports (88 vs. 82%; W = 112.5; *p* = 0.008; r = 0.34; Figure 5b). The maximum length of the circumnutation was wider for the ‘Control-Thin’ than for the ‘Control-Thick’ condition (135 vs. 113 mm; W = 203; *p* = 0.018; r = 0.30). The finding is important because it confirms that *P. sativum* plants exhibit a different kinematic pattern for thick and thin supports [20,22,23] and it means that this patterning is instrumental for investigating the effects determined by the perturbed conditions. No significant differences were found in the maximum duration of the circumnutation for the thinner than the thicker supports (96 vs. 105 min; W = 296; *p =* 0.615; r = 0.06).

#### 3.2.2. ‘Thin-Below’ vs. ‘Control-Thick’

The movement time was longer for the perturbed than for the control condition (2592 vs. 1638 min; W = 148.5; *p* < 0.001; r = 0.59; Figure 4), and the maximum velocity of the tendrils was higher for the perturbed than for the control condition (15 vs. 12 mm/min; W = 192; *p* < 0.001; r = 0.43; Figure 6a). The time when the maximum velocity of the tendrils occurred was earlier for the perturbed than for the control conditions (46 vs. 67%; W = 482; *p* = 0.040; r = 0.26; Figure 6a). The maximum aperture of the tendrils was greater for the perturbed than for the control condition (53 vs. 30 mm; W = 126; *p* < 0.001; r = 0.53; Figure 6b). Further, the time at which the maximum aperture of the tendrils occurred was earlier for the perturbed than for the control trials (69 vs. 82%; W = 417; *p* = 0.021; r = 0.30; Figure 6b). The maximum length of the circumnutation was wider for the ‘Thin-Below’ than for the ‘Control-Thick’ (149 vs. 113 mm; W = 200; *p* = 0.001; r = 0.42). The maximum duration of the circumnutation was longer for the ‘Thin-Below’ than for the ‘Control-Thick’ support (153 vs. 105 min; W = 118; *p* < 0.001; r = 0.58).

#### 3.2.3. ‘Thick-Below’ vs. ‘Control-Thin’

The movement time was longer for the perturbed than for the control condition (2350 vs. 1145 min; W = 216; *p* < 0.001; r = 0.47; Figure 4). Similarly, the peak of the maximum velocity of the tendrils occurred earlier for the perturbed than for the control trials (55 vs. 81%; W = 470; *p* = 0.006; r = 0.35; Figure 6b). The maximum duration of the circumnutation was longer for the ‘Thick-Below’ than for the ‘Control-Thin’ support (165 vs. 96 min; W = 41; *p* < 0.001; r = 0.71). No significant differences in the maximum velocity of the tendrils (14 vs. 14 mm/min; W = 346; *p* = 0.413; r = 0.10; Figure 6c), the aperture (43 vs. 39 mm; W = 247; *p* = 0.422; r = 0.10; Figure 6d), the peak of the maximum aperture of the tendrils (85 vs. 88%; W = 284; *p* = 0.143; r = 0.19; Figure 6d), and the maximum length of the circumnutation (130 vs. 135 mm; W = 333; *p* = 0.507; r = 0.08) were found for the ‘Thick-Below’ as opposed to the ‘Control-Thin’ conditions. 

## 4. Discussion

The study explored the interaction between the below- and above-ground parts of *P. sativum* plants for the purpose of coding information about the thickness of a possible support. The results showed that there were differences in the kinematic patterning when the perturbed and control trials were compared. First and foremost, the movement duration for the perturbed trials was longer than for the control trials, suggesting that the thickness mismatch affects the duration of the movement. The supports whose lower and upper parts were different probably require more processing with respect to a single diameter support because more information needs to be processed. Longer processing times were also evident as far as the maximum duration of the circumnutation pattern was concerned. For the perturbed trials, the maximum duration of the circumnutation was longer than for the control trials. Hypothetically the plant needs more time for the circumnutation process when two sizes need to be analyzed. 

When the ‘Thin-Below’ with the ‘Control-Thick’ conditions were compared, we found that the results depended on the kinematic measures considered. The amplitude of the maximum velocity and the maximum aperture of the tendrils suggest that *P. sativum* plants adjust the kinematic pattern of the movement depending on the below-ground (i.e., thin) part of the support. In other words, the maximum velocity of the tendrils was higher, and their maximum aperture was wider for the perturbed trial than for the ‘Control-Thick’ condition. It is to be remembered that this is the pattern that was previously observed for all thinner single-sized supports [20,22,23]. Furthermore, the maximum length of the circumnutation was wider for the ‘Thin-Below’ than for the ‘Control-Thick’ conditions. This pattern mirrors the one observed when the control conditions were compared. As previously reported, reaching and grasping a thick support (i.e., those with larger diameters) appears to be more ‘difficult’ for a climber with respect to grasping a thinner one because it is more energy consuming. Indeed, the plant not only needs to increase the length of its tendrils to efficiently wrap itself around the support, but it also needs to strengthen tensional forces to counteract gravity [3,48]. Given these considerations, it would seem that *P. sativum* plants need to contain the amplitude of the circumnutation movements in order to save energy to improve their efficacy [20,21]. Overall, the data gathered infer that the pattern of the movement as far as amplitudes are concerned is based on the information provided by the root system. However, with regard to the temporal parameters, it was found that the times at which the maximum peak velocity and the peak grip aperture of the tendrils occurred was earlier for the perturbed than for the control condition. While one could reason that there should be no differences because the size of the aerial part of the ‘perturbed’ support and the size of the control support are similar, it is possible that the higher velocities linked to processing the ‘thin’ part of the support lead to an earlier occurrence of key kinematic parameters in order to establish a functional equilibrium between the information processed at the two levels (the aerial and underground parts). In other words, the times of the maximum velocity and aperture of the tendrils are crucial landmarks because they reflect, respectively, the moment when the tendrils start to slow down and begin to lean towards the support during the approaching phase. It is a critical time as the plant is actively latching onto a possible support. If the ‘time’ information regarding the underground part of the support is inappropriate to maximize the possibility of attachment, then timing modifications would need to be made by the aerial organs of the plant. Altogether these observations suggest that there is a crosstalk between the roots and the aerial components of the plant, something that is evident at a kinematic level.

As far as the comparison between the ‘Thick-Below’ and the ‘Control-Thin’ conditions is concerned, we found no evidence for any kind of crosstalk and no kinematic effects linked to the perturbation condition. In these cases, it would seem that what has been programmed on the basis of the information coming from the underground part of the support (i.e., the thick part) fits the requirements needed for grasping the above part (i.e., thin). We are tempted to interpret these findings in connection to the demonstration that for *P. sativum* plants [20,22,23] and climbing plants in general [13] grasping a thicker support is a more demanding activity than grasping a thinner one. Therefore, it might be easier to adapt a pattern of movement related for grasping a thicker more demanding support for grasping a less demanding thinner support. In such circumstances, the effects of the perturbation are minimized and no difference with the control condition is found.

Overall, these results indicate that the roots convey ‘information’ to the plant that is used to regulate its growth and behavior, accordingly; they also suggest that there is a sort of functional equilibrium between the roots and aerial organs via an interactive signaling process [45,49,50]. Other studies have in fact reported that root systems have indirect effects on functional traits and attributes including, for instance, leaf size and the capacity to orient leaves. Ultimately then, it would seem that the root system does indeed affect the plant’s growth regulation and activity [49]. The results of this work expand the scope of these indirect effects to the tendrils’ ability to adapt their movement to the thickness of their potential supports. They also open the door to the possibility that there is a crosstalk between the roots and the aerial parts as a response to fluctuating environmental conditions. 

## 5. Conclusions

These results suggest that a crosstalk exists between the above- and the below-ground parts of *P. sativum* plants for the purpose of processing information about the thickness characterizing a potential support. The integration of information from both above- and below-ground organs could permit a plant to detect and respond to environmental factors surrounding it and to flexibly adapt its behavior to ever-changing situations [24,36,38,51]. Future studies aiming to investigate both kinematic and physiological measures are needed to further explore the functional equilibrium and interactivity between plants’ above- and below-ground organs. An integrated kinematic and physiological analysis of plant growth responses will undoubtedly expand our understanding of plant behavior and ecophysiology.

## Figures and Tables

**Figure 1 biology-11-00405-f001:**
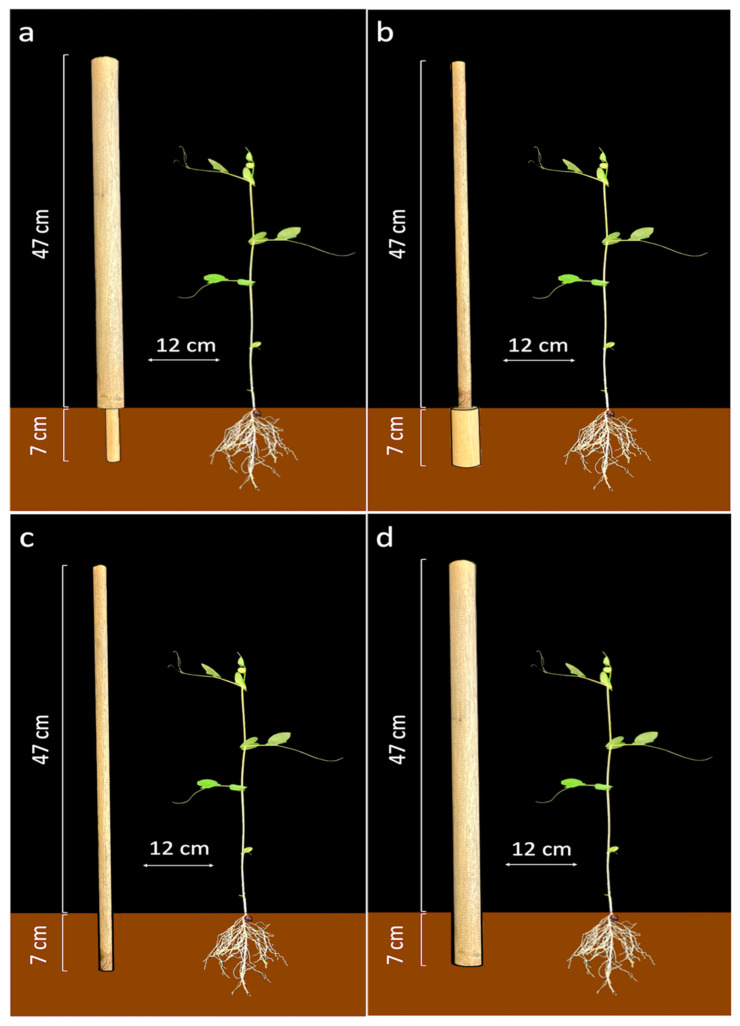
Supports for the: (**a**) ‘Thin-Below’ condition, (**b**) the ‘Thick-Below’ condition, (**c**) the ‘Control-Thin’ condition, and (**d**) the ‘Control-Thick’ condition. In each case the support was positioned in front of the first plant’s leaf at a distance of 12 cm.

**Figure 2 biology-11-00405-f002:**
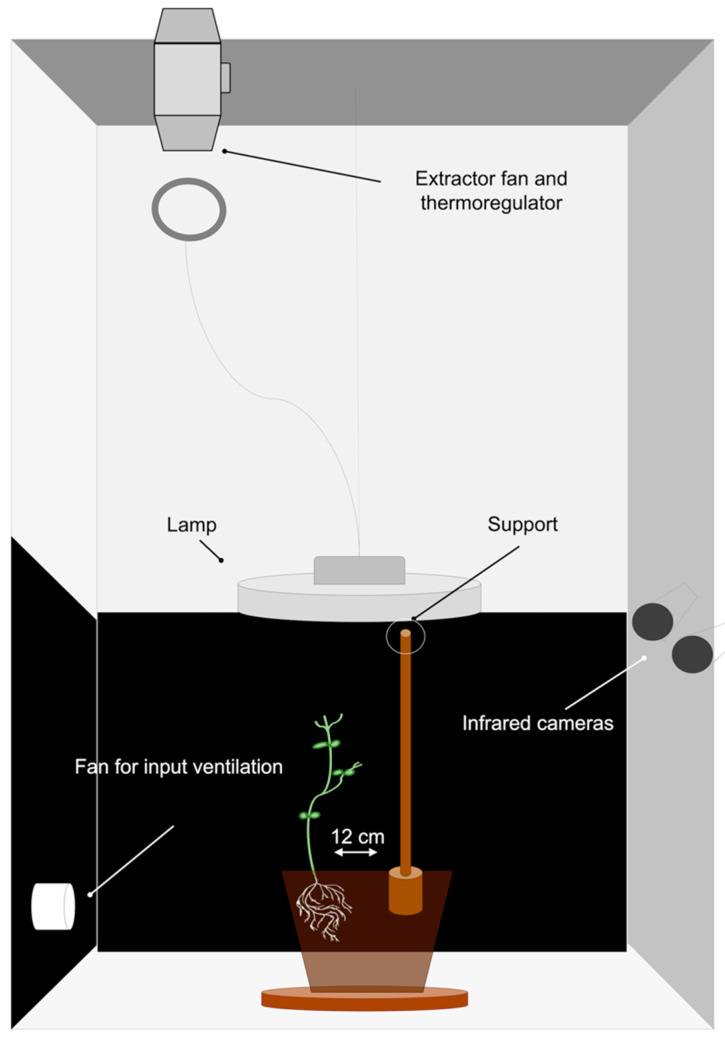
Graphical representation of the experimental set up. The representation illustrates the ‘Thick-Below’ condition.

**Figure 3 biology-11-00405-f003:**
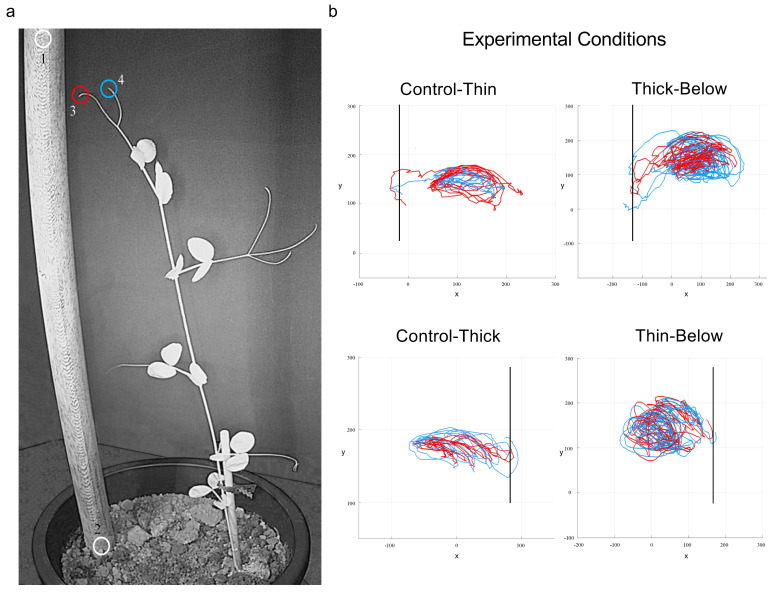
The landmarks considered and examples of the spatial trajectories. (**a**) The landmarks considered were the support (1, 2) and the tip of the tendrils (3, 4). The colors of the circles correspond to the colors of the trajectories shown in the right-side panel (i.e., light-blue, and red lines in panel b refer to the individual trajectory for each tendril). (**b**) represents the trajectories for the tip of the tendrils for the ‘Control-Thin’, ‘Thin-Below’, ‘Control-Thick’ and ‘Thick-Below’ support conditions. The support is a solid vertical line. The axes x and y refer to the sagittal and vertical axis in mm, respectively. Please note that the plant leaned toward the support and the tendrils grasped it for all the conditions.

**Figure 4 biology-11-00405-f004:**
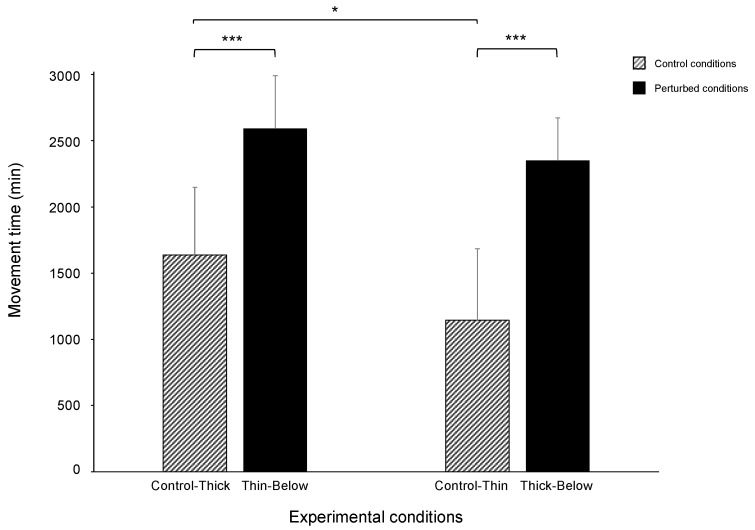
Bar plots representing movement time during the ‘Control-Thick’ and ‘Thin-Below’ support conditions and during the ‘Control-Thin’ and ‘Thick-Below’ support conditions. The bars refer to the median; the error bars refer to the absolute deviation (MAD). The asterisks indicate when the difference between the perturbed and the control conditions was significant. Please note that movement duration was longer for the ‘Control-Thick’ than for the ‘Control-Thin’ conditions and they were longer for the perturbed (i.e., ‘Thin-Below and ‘Thick-Below’ support) than for the control conditions. * = *p* < 0.05; *** = *p* < 0.001.

**Figure 5 biology-11-00405-f005:**
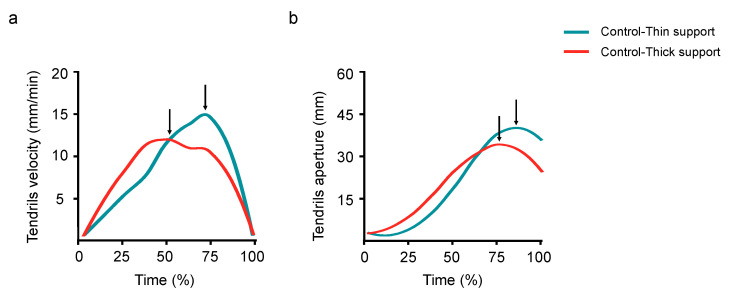
A graphical representation of the velocity of the tendrils (**a**) and of their aperture (**b**) for the ‘Control-Thin’ (i.e., the light blue line) and the ‘Control-Thick’ conditions (i.e., the red line). The arrows indicate the moments when the maximum peak velocity (**a**) and the maximum peak aperture (**b**) occurred. Please note that these peaks occurred earlier for the ‘Control-Thick’ than for the ‘Control-Thin’ conditions.

**Figure 6 biology-11-00405-f006:**
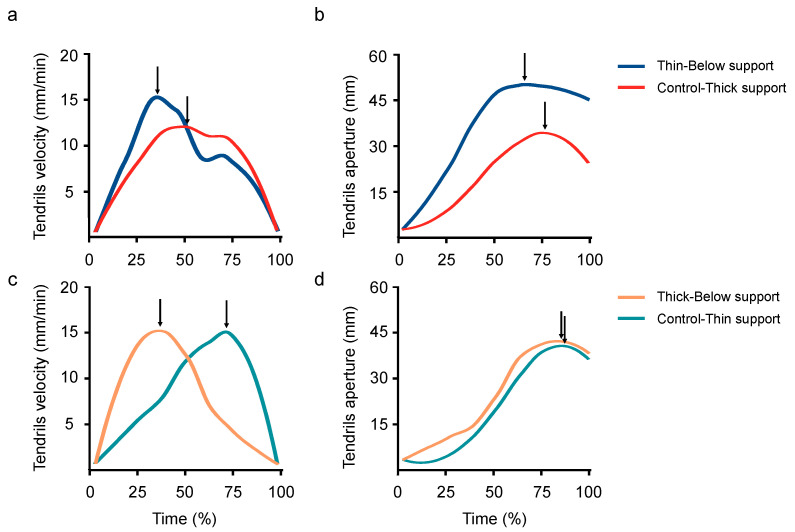
A graphical representation of the velocity (**a**,**c**) and the aperture of the tendrils (**b**,**d**) during the ‘Control-Thick’ (i.e., the red line) and the ‘Thin-Below’ conditions (i.e., the blue line) and during the ‘Control-Thin’ (i.e., the light blue line) and the Thick-Below conditions (i.e., the orange line). The arrows indicate the time the maximum peak velocity (**a**,**c**) and maximum peak aperture of the tendrils (**b**,**d**) occurred. Please note that for the perturbed trials (i.e., ‘Thin- and ‘Thick-Below’ condition) the peak velocity of the tendrils was reached earlier with respect to what happened during the control trials (i.e., ‘Control-Thin’ and ‘Control-Thick’ conditions; (**a**,**c**)). Further, the peak of the maximum aperture of the tendrils occurred earlier for the ‘Thin-Below’ than for the ‘Control-Thick’ support (**b**). No differences were found in the maximum aperture of the tendrils and in the peak of the maximum aperture for the ‘Control-Thin’ and ‘Thick-Below’ conditions (**d**).

**Table 1 biology-11-00405-t001:** Sample description.

	**Control-Thick vs. Thin-Below**	
	Control-Thick	Thin-Below
N°	10	10
Germination period	6 d (±0.5; Range 4–10)	5 d (±1.22; Range 5–12)
Age	21 d (±3.1; Range 14–26)	16.5 d (±1.7; Range 14–19)
	**Control-Thin vs. Thick-Below**	
	Control-Thin	Thick-Below
N°	10	10
Germination period	5.5 d (±0.6; Range 4–7)	5 d (±1.5; Range 3–10)
Age	14 d (±2; Range 10–20)	21.5 d (±5.6; Range 9–26)

Note. The germination period and age, which are expressed in days, refer to the median, while the median absolute deviation is reported in the parentheses. The age of the plants corresponds to the period of time starting from the germination of the seed to the grasping of the support by the plant. d = days.

## Data Availability

The data set used and analyzed during the current study is available from the corresponding author upon request.

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
