# Peer review of "Kinematic Evidence of Root-to-Shoot Signaling for the Coding of Support Thickness in Pea Plants"

_biology, 2022, doi:10.3390/biology11030405_

Round 1
Reviewer 1 Report
I would like to thank authors for considering the comments and improving their manuscript accordingly.
Author Response
We thank the Reviewer for finding the time to evaluate the new version of our manuscript. We are very pleased that he/she found our response to the raised issues satisfactory. Indeed his/her comments have been invaluable to improve the quality of the manuscript.
Reviewer 2 Report
The manuscript entitled “Kinematical evidence of root-to-shoot signalling for the coding of support thickness in pea plants” mainly reports on possible shoot-root interactions that could influence the ability of a climbing plant (Pisum sativum L.) to locate a support in the environment.
The manuscript is well written; the experiments are well described and performed; the topic is relevant for Biology’s readers.
My specific comments are reported below.
Line 115. Authors should avoid interpreting some plant sensory mechanisms as “plant-specific vision systems”. These systems typically involve the reception of light and the capacity to build a representation of the surrounding environment.
Line 193. Table 1. The authors compared plants of different ages without justification in the text. Why? Could the plant age influence the observed behaviors?
Author Response
We thank the Reviewer for finding the time to evaluate the new version of our manuscript. Indeed his/her original and present comments have been invaluable to improve the quality of the manuscript.
1. Line 115. Authors should avoid interpreting some plant sensory mechanisms as “plant-specific vision systems”. These systems typically involve the reception of light and the capacity to build a representation of the surrounding environment.
R1. We thank the Reviewer for this clarification, the text has been changed accordingly. Please refer to lines 120-124.
2. Line 193. Table 1. The authors compared plants of different ages without justification in the text. Why? Could the plant age influence the observed behaviors?
R2. We would like to thank the Reviewer for giving us the possibility to clarify this aspect that has now been mentioned within the text. The age of the plants corresponds to the period of time starting from the germination of the seed to the grasping of the support by the plant. Naturally this time varied depending on the pattern of growth exhibited by the plants.
Concerning the relationship between the plant age and the observed kinematical patterning to our knowledge there are no reports on this issue. Jaffe and Galston (1966) reported that tendrils of the Alaska pea plant (P. sativum L.) exhibited some differences in terms of irritability at different ages, but this aspect is outside the scope of the present manuscript. Having said that the Reviewer is raising an interesting issue that we have pursued comparing the kinematical patterning of plants differing in age within each experimental condition. We found no differences for the considered dependent measures depending on age.
Reviewer 3 Report
This study showed that information about different thickness of support conveyed by the root system play important role to fulfill the end-goal of the tendril movement. The study was easy to follow, the results are interesting and thought-provoking. The manuscript is well written with good structure and logical flow. The results of this research can contribute to better understanding of within plant signaling and their role in plant behavior in specific environment.
However, I have some concerns. My biggest concern about this research is how the authors can prove that the roots came into contact with support. How the root determined the thickness of support? Small stick used to support the plant (Figure 3a) could also be detected by roots and thus bias experiment. How the root distinguished between these two sticks? This needs to be clarified in M&M.
According to originality report the manuscript has similarity index of 29% and while only with the article “Silvia Guerra, Bianca Bonato, Qiuran Wang, Francesco Ceccarini et al. "The coding of object thickness in plants: When roots matter." the similarity index is 16%. I think this should be corrected before publishing.
I have some suggested changes which are relatively minor and should be easy to implement.
Table 1: I am just curious is there any relation between plant age and tendril movement.
Line 224: “5.45 am to 5 pm” gives 11 hours and 15 minutes
Author Response
We thank the Reviewer for finding the time to evaluate the new version of our manuscript. Indeed his/her original and present comments have been invaluable to improve the quality of the manuscript.
1. My biggest concern about this research is how the authors can prove that the roots came into contact with support. How the root determined the thickness of support? Small stick used to support the plant (Figure 3a) could also be detected by roots and thus bias experiment. How the root distinguished between these two sticks? This needs to be clarified in M&M.
R1. Unfortunately, we cannot prove that the roots met the support. From the present data the fact that the root system is involved in the coding of stimulus thickness can only be inferred on the basis of plants’ response to our experimental manipulations and at speculative level on the basis of previous studies. As we report within the introduction section, the root system has the ability to perceive and process different stimulus properties by touching the inground part of the support (Darwin & Darwin, 1880; Del Bianco & Kepinski, 2021) or by means of root exudates (Massa & Gilroy, 2003; Semchenko et al., 2008).
Concerning the presence of the small stick showed in Figure 3, we thank the Reviewer for picking up this aspect that has now been clarified within the ‘methods’ section. The stick was applied to sustain the plant after germination. We were aware of a potential problem if the stick would have been perceived by the roots, therefore we built it as an L shape and fixed to the pot’s rim. The part running from the vertical component of the stick to the pot’s rim was not in grounded but simply covered by a thin layer of silica sand.
2. According to originality report the manuscript has similarity index of 29% and while only with the article “Silvia Guerra, Bianca Bonato, Qiuran Wang, Francesco Ceccarini et al. "The coding of object thickness in plants: When roots matter." the similarity index is 16%. I think this should be corrected before publishing.
R2. As kindly requested, this issue has been fixed. We have changed the text so to avoid similarities with our previous manuscript. The similarity index is now 4% and chiefly refers to unavoidable methodological details (COMPILATIO.NET).
3. Table 1: I am just curious is there any relation between plant age and tendril movement.
R3. Regarding the relationship between the plant age and the observed kinematical patterning to our knowledge there are no reports on this issue. Jaffe and Galston (1966) reported that tendrils of the Alaska pea plant (P. sativum L.) exhibit some differences in terms of irritability at different ages, but this aspect is outside the scope of the present manuscript. Having said that the Reviewer is raising an interesting issue that we have pursued comparing the kinematical patterning of plants differing in age within each experimental condition. We found no differences for the considered dependent measures depending on age.
4. Line 224: “5.45 am to 5 pm” gives 11 hours and 15 minutes
R4. We apologize with the Reviewer for this mistake. We have now revised the manuscript accordingly (see line 242).
This manuscript is a resubmission of an earlier submission. The following is a list of the peer review reports and author responses from that submission.
Round 1
Reviewer 1 Report
The manuscript entitled “Root-to-shoot signalling for the coding of support thickness in pea plants” represents a physiology trial using the simply well-designed 3D kinematical model under comparative “control-thick and thin-below support” conditions in Pisum sativum L., to verify three speculated puzzles: 1. The root senses the support thickness 2. Shoot involved in sensing the support thickness leading to no difference in perturbed support 3. Crosstalk between shoot and root may cause variation by below and above ground supports. This is a well-written manuscript, especially for the introduction and discussion, providing more recent advances with a particular focus on the climbing plant's behavior and the underlying biological and physiological mechanisms.
However, owing to the same system used in this work, a vast proportion of data are overlapped with their previous work (check the cited reference s38-40), prompting the lack of significance and novelty. The authors need to provide even more sufficient and convincing evidence to code the biological function of the support thickness under the complicated “root-to-shoot signalling” regime. The authors addressed too many theories and speculations, but very few of them supported the current data and results. For instance, Figure 4 is the ONLY readable value from this work, but the result is terrible because the shoot movement times in two different controls were insignificantly variable (this makes no sense and is inconsistent with the addressed findings in the introduction). Besides, the results and respected conclusion presented were sketches and vague, meaning that the experiment should be re-conducted urgently integrating with physiological measurement (as the author claimed in the conclusion part) and stastic analyses. Thus, I strongly recommend this manuscript for rejection. Authors should reconsider resubmission until the solid date is provided appropriately.
Some of a few formal comments on the manuscript as following indicated:
- Pls remove the redundant sentences unrelated to the results and speculation; reduce too many hypotheses and the number of references cited.
- In this work, substantial revision should be conducted in materials and methods, and how many replicates and biological repeats are performed. Specify the descriptions like“Healthy-looking pea seeds...” etc.
- A detailed legend should be provided in each figure and table.
Reviewer 2 Report
Manuscript of Guerra et al., aim to unravel the root-to-shoot signaling for support thickness detection in Pisum sativum. Authors performed the experiment of observing the growth and movement of tendrils by automated system. Overall conclusion is that there is some communication between root and shoot.
However, authors did not provide any other results or evidences. The whole manuscript consist of just one experiment and discussing what the potential mechanism might be. Unfortunately, nothing else was evaluated or tested and with this amount of data, authors did not provide enough for this manuscript to be positively evaluated in this journal.
Reviewer 3 Report
none needed